# Ultramarathon Plasma Metabolomics: Phosphatidylcholine Levels Associated with Running Performance

**DOI:** 10.3390/sports8040044

**Published:** 2020-04-01

**Authors:** Tracy B. Høeg, Kenneth Chmiel, Alexandra E. Warrick, Sandra L. Taylor, Robert H. Weiss

**Affiliations:** 1Department of Physical Medicine & Rehabilitation, University of California, Sacramento, CA 95817, USA; aewarrick@ucdavis.edu; 2Mountain View Rehabilitation Medical Associates, Grass Valley, CA 95945, USA; 3Napa Medical Research Foundation, Napa, CA 94558, USA; 4Division of Nephrology, Department of Medicine, University of California, Davis, CA 95817, USA; kjchmiel@ucdavis.edu (K.C.); rhweiss@ucdavis.edu (R.H.W.); 5Division of Biostatistics, Department of Public Health Sciences, University of California, Davis, CA 95616, USA; sltaylor@ucdavis.edu; 6Medical Service, VA Northern California Health Care System, Sacramento, CA 95655, USA

**Keywords:** exercise, physiology, endurance, running, ultramarathon, performance

## Abstract

The purpose of this study was to identify plasma metabolites associated with superior endurance running performance. In 2016, participants at the Western States Endurance Run (WSER), a 100-mile (161-km) foot race, underwent non-targeted metabolomic testing of their post-race plasma. Metabolites associated with faster finish times were identified. Based on these results, runners at the 2017 WSER underwent targeted metabolomics testing, including lipidomics and choline levels. The 2017 participants’ plasma metabolites were correlated with finish times and compared with non-athletic controls. In 2016, 427 known molecules were detected using non-targeted metabolomics. Four compounds, all phosphatidylcholines (PCs) were associated with finish time (False Discovery Rate (FDR) < 0.05). All were higher in faster finishers. In 2017, using targeted PC analysis, multiple PCs, measured pre- and post-race, were higher in faster finishers (FDR < 0.05). The majority of PCs was noted to be higher in runners (both pre- and post-race) than in controls (FDR < 0.05). Runners had higher choline levels pre-race compared to controls (*p* < 0.0001), but choline level did not differ significantly from controls post-race (*p* = 0.129). Choline levels decreased between the start and the finish of the race (*p* < 0.0001). Faster finishers had lower choline levels than slower finishers at the race finish (*p* = 0.028).

## 1. Introduction

A low level of cardiorespiratory fitness (CRF) has been associated with higher all-cause mortality and specifically with increased risk of cardiovascular disease and numerous cancers [1]. There is significant evidence that CRF is a more powerful predictor of mortality than obesity, hypertension, smoking and type 2 diabetes [1]. Elite ultramarathon runners have one of the highest known levels of CRF, being capable of running 161 km on mountainous terrain in under 24 h. Distinguishing the metabolomic profile of these elite athletes from slower runners and sedentary adult controls may provide useful insight into the biological processes associated with increasing levels of fitness, decreased mortality and decreased risk of many diseases. 

Metabolomics identifies the final endpoints of biochemical processes and closely reflects an organism’s phenotype, environment and behavior [2]. Exercise metabolomics can provide valuable insights regarding the end biological impact of repeat and prolonged physical activity [3] as well of the biological processes associated with higher levels of endurance performance. The metabolomic profile associated with superior endurance sports performance has yet to be explored, though metabolomics studies have already identified choline-containing compounds and certain phospholipids to be positively associated with level of physical fitness [4,5]. Numerous studies have looked at the immediate impact of exercise on the metabolome [6,7,8,9], however there is little information on the difference in the metabolomic profiles of exceptionally physically fit vs. sedentary adults. Studying elite ultramarathon runners vs. non-athletic controls allow us to make the unique comparison of two groups with starkly different levels of physical fitness and add to the current limited literature available regarding the metabolites associated with increased physical fitness.

The Western States Endurance Run (WSER) is a 161-km foot race, which takes place on a dirt trail from Squaw Valley to Auburn, California, USA, the 4th Saturday of June every year. The race has over 12,000 m in overall elevation change and temperatures can often exceed 40 °C. The unique endurance demands and organization of this ultramarathon, including participation of both very fast and relatively slower runners served as an ideal setting for this study. In Phase I of this study, we hypothesized that non-targeted plasma metabolomics would identify specific metabolites or groups of metabolites to be associated with a faster ultramarathon finish time. Phosphatidylcholines (PCs) emerged as being associated with ultramarathon performance. This was consistent with the previous literature noting PCs to be positively correlated with VO2max [4] and inversely correlated to neuromuscular disease severity [10]. In Phase II of the study, using targeted plasma metabolomics, we hypothesized that pre- and post-race levels of PCs would be 1. Associated with faster ultramarathon finish time, 2. Related to plasma choline levels and, 3. Higher in runners than in the non-athletic controls. 

## 2. Materials and Methods

This study was approved by the Institutional Review Board at UC Davis (#824889). Research subjects were registered runners for the WSER and were recruited via a pre-race informational email prior to their participation in 2016 and/or 2017. All runners were invited to participate and all interested in participating signed a pre-race consent form. Enrollment was limited to 50 participants in 2016 and 52 in 2017. The exclusion criteria were age <18 years, pregnancy, and inability to read and/or speak English. A participant flow chart is shown in Figure 1. 

### 2.1. Participant Flow

In Phase I in 2016, runners who consented to take part in the study prior to the race had their blood drawn within an hour of finishing the race. Runners who did not finish the race, refused the post-race blood draw, or in whom blood could not be obtained, were excluded from the study. Non-targeted metabolomics analysis was performed on all participants’ samples. 

In Phase II in 2017, runners who consented to take part in the study, had a pre-race blood sample one day before the race and, then again post-race, within one hour of finishing. Runners who did not finish the race, refused the post-race blood draw, or in whom blood could not be obtained were excluded from the study. Targeted lipidomics and choline analysis was performed on all participants’ pre- and post-race samples.

### 2.2. Performance Endpoint

161-km race finish time was chosen as our performance endpoint, as a proxy for CRF. 

### 2.3. Metabolomic Analysis

All blood was drawn by the same phlebotomists in a given year using identical technique. The plasma, obtained by centrifuging whole blood at room temperature, was immediately placed on dry ice in the field and the next day moved to a laboratory freezer kept at −80 °C until the metabolomics analysis, which occurred within several months. In 2016, approximately half of each plasma sample underwent mass-spectrometry based non-targeted metabolomic analysis [11]. Non-targeted metabolomic profiles were obtained using CSH-QTOF MS/MS in both positive and negative modes as previously reported [11]. The other half remained frozen at −80 °C. In 2017, the same protocol was followed, except that targeted metabolomic analysis was performed for phospholipids and choline (based on the 2016 results). All samples were quantified for amount of hemolysis using the Mayo Clinic’s “Color Chart for Detection of Hemolysis” [10]. 

### 2.4. Control Subjects

In Phase II, fourteen adult (≥30 years) sex-matched and age range-matched control plasma samples for the 2017 participants from self-identified “non-athletic” donors were purchased from, BioreclamationIVT (Westbury, NY, USA). We had one control for every two participant samples. The control samples were also submitted for targeted lipidomics and choline analysis as described above.

### 2.5. Statistical Analysis 

For the 2016 data, a two-sample t-test was used to compare the post-race GCTOF (gas chromatography time of flight) mass spectrometry and lipid values of our 10 fastest finishing with our 10 slowest finishing participants. False discovery rates (FDR) were calculated for each analysis to account for multiple testing. Statistical testing was two-sided and considered significant if FDR < 0.05. 

For the 2017 data, paired sample t-tests were used to test for difference between pre- and post-race choline and PC levels in runners. Two-sample t-tests were used to compare pre- and post-race choline and PC levels in runners to the non-athletic controls; PC levels were log transformed for this analysis to meet assumptions of normality and homogeneity of variances. Finish time was related to pre- and post-race choline levels using simple linear regression. Prior to analysis, using all metabolites, the intensity values of each sample were total quantity normalized to the median total intensity level. Only positive mode PC levels were analyzed in Phase II of the study, as positively charged PCs are subject to a high amount of electrospray depression in the negative mode. All statistical tests were two-sided and considered significant if FDR < 0.05. 

## 3. Results

### 3.1. Participant and Control Characteristics

Phase I of the study was conducted at the 2016 WSER and plasma samples were obtained post-race from 39 of the 49 consented runners (Figure 1), 8/39 (20.5%) being female. Mean age was 43 years (95% CI = 40–46, range = 29–66). All runners who consented to participate pre-race were considered “consented runners”. Nine runners did not finish the race and one opted to not have blood drawn after multiple failed attempts; these individuals were excluded from the study.

In Phase II of the study, conducted at the 2017 WSER, plasma samples were obtained post-race from 28 of the 52 consented runners (Figure 1), with 4/28 (14.3%) being female. Mean age was 41 years [95% CI = 37–45, range = 27–59]. One of 28 pre-race plasma samples could not be obtained due to multiple failed phlebotomy attempts. Seventeen runners did not finish the race and an additional 7 either did not present to the finish line tent to have their blood drawn or phlebotomy was unsuccessful. Fourteen control samples (Figure 1) of which 2 (14.3%) were female. Mean age was 50 (95% CI 44–56, range = 30–69).

### 3.2. Phase I: Non-targeted Metabolomics

In 2016, non-targeted metabolomics analysis was performed using post-race serum from the 39 participants. Of 427 known compounds detected, four were significantly associated with finish time (FDR < 0.05) with these being PCs 40:8, 38:5 A, o-32:0 and PC 40:4 or 40:5 (Figure 2). 

### 3.3. Phase II: Targeted Lipidomics

In 2017, based on the above results from 2016, we performed targeted lipidomics of PCs in pre-and post-race serum of participating ultramarathon runners and a control, “non-athletic” group. 

#### 3.3.1. PC Levels and 161-km Finish Time

In targeted lipidomics of PCs in runners’ pre-race plasma, two PCs (PCs 42:4 or 42:5 and PC 44:4 or 44:5) (Figure 3) were found to be significantly higher in faster finishers (FDR < 0.05). In targeted lipidomics of PCs in runner’s post-race plasma, four PCs (37:4, 38:7, 40:6A, and 42:5A) were found to be significantly higher in faster finishers (FDR < 0.05) (Figure 3). None of these specific PCs corresponded with the PCs significantly correlated in 2016. 53/85 PCs were found to significantly change (FDR < 0.05) in level over the course of the race with 8/85 (9.4%) decreased in level and 45/85 (52.9%) increased. 

#### 3.3.2. PC Levels in Race Participants vs. Non-athletic Controls

Targeted lipidomics of PCs in runners’ pre-race plasma were compared with the levels of non-athletic controls (Figure 4). 41 were found to be significantly higher in runners than controls (FDR < 0.05) and four were found to be significantly lower (FDR < 0.05). Targeted lipidomics of PCs in runners’ post-race plasma were compared with the levels of non-athletic controls (Figure 4). 61 were found to be significantly higher in the runners than the controls (FDR < 0.05) and 2 were found to be significantly lower (FDR < 0.05). All four PCs identified in 2016 as being associated with faster finish time were found to be elevated in participants vs. controls either from pre-race or post-race samples or both. 5/6 PCs identified in 2017 as being associated with finish time were also elevated in participants vs. controls again either pre-race, post-race or both. 

### 3.4. Phase II: Choline 

Choline in runners’ plasma was found to decrease significantly from pre- to post-race (*p* < 0.0001); the mean ± SD for choline in pre-race samples was 289,252 ± 74,430 counts/s and post-race samples 154,449 ± 28,371 counts/s. Pre-race runners’ choline differed significantly from controls (t = −9.03); the mean ± SD for choline in controls was 138,504 ± 32,119 counts/s. Post-race runners’ choline did not differ significantly from controls (*p* = 0.129). 

Additionally, faster finish time was associated with lower levels of post-race plasma choline (*p* = 0.028). Finish time was not significantly related to pre-race choline levels (*p* = 0.539). 

## 4. Discussion

The most important finding of this study is the apparent relationship between faster ultramarathon finish time and higher levels of certain phosphatidylcholines. This was found both in the non-targeted metabolomics portion of this study and subsequently using targeted metabolomics in a different group of runners the following year. Furthermore, ultramarathon runner participants had significantly higher PC levels than non-athletic controls for most of the PCs measured. These findings are consistent with those of a study by Bye et al. [4], which found VO2max to be highly correlated with PC levels using whole blood lipidomic analysis. These findings are also consistent with Mastrokolias et al. [10], who found lower PC levels in a chronic neuromuscular disease population, with the lowest levels among those with greater disability. Newsom et al. [12] found the PC levels in skeletal muscle to be significantly higher in athletes than in participants with obesity or with type 2 diabetes. Additional studies have supported these findings with an inverse relationship between PC levels and obesity [13] and type 2 diabetes [14]. These findings taken together suggest a relationship between PC levels and regular exercise, physical fitness and/or body composition. 

One could hypothesize that the elevated PC levels in athletes is secondary to cellular membrane breakdown from physical exertion. PCs are indeed major components of biological cell membranes. However, though the majority of the PCs we identified significantly increased, a number of others significantly decreased over the course of the 161-km race. Additionally, we found both pre-and post-race PC levels to be higher in athletes. Consistent with this, Newsom et al. [12] found elevated PC levels in skeletal muscle samples both before and after exercise. Additionally, it would be unexpected for the most physically fit athletes to have more muscle breakdown than the less physically fit when running the same distance. These findings together suggest most of the elevation in PCs is chronic rather than acutely related to a bout of exercise.

From a practical standpoint, the metabolites and metabolomic pathways associated with CRF are more likely to represent the phenotype of superior cardiovascular and/or musculoskeletal fitness than those associated with a single bout of exercise. The rise in phosphatidylcholine molecules chronically in physically fit individuals, as was shown in our study and numerous other studies [4,10,13,14], indicates that the PC pathway either plays a critical role in physical fitness or is highly associated with another critical pathway that does. As we attempt to gain insight into the molecular mechanisms behind the health benefits of exercise, the PC pathway deserves further investigation.

Our study also analyzed the relationship between choline and PC levels as well as choline and ultramarathon performance. Choline is required for the synthesis of PCs [14,15] and choline supplementation has been found to improve endurance running performance [16,17]. We did find, as expected [15,18,19], that choline levels decreased from pre-to post-race. However, we did not find pre-race choline levels to be related to finish time. Interestingly, faster finish time was associated with significantly lower post-race choline levels. This would suggest a relationship between physical exertion and choline usage, as has been found in marathon running [18]. 

Systemic availability of choline appears to be a limiting factor in athletic performance [15,16]. The elevated plasma PC levels pre-race may be due to long-term hepatic adaptation driven by frequent choline depletion and the need to replenish acetylcholine at the neuromuscular junction. PCs may be chronically elevated to provide an additional source of choline, and thus acetylcholine, for use during exercise. Approximately 95% of the body’s choline is stored as phosphatidylcholine [20]. Choline is obtained primarily either directly through the diet or through conversion of phosphatidylcholine to choline through the phosphatidylethanolamine *N*-methyltransferase (PEMT) pathway [20]. Li et al. [21] found elevated choline to be associated with lower PC levels and decreased physical fitness. 

Bye et al. [4] suggest that phospholipase D (PLD), a plasma membrane enzyme responsible for the hydrolysis of PC to choline, may be more active in sedentary individuals than athletes. This would result in higher choline and lower PCs in sedentary adults, and the reverse in athletes, which is generally consistent with our results. However, we did unexpectedly find significantly increased choline concentrations pre-race in athletes compared to the controls. This may be due to increased dietary choline availability before the race; indeed, fasting has been shown to decrease plasma choline levels [7]. Our participants were likely to be in a well-fed state prior to an ultramarathon. Unlike our study, the study by Bye et al. [4] required fasting of both participants and controls. 

There are several limitations of this study. First, due to the limited size of the ultramarathon participant pool and available funding, we had a relatively small participant number for both phases of the study. This small number of participants increased our risk of a Type II statistical error, or not detecting significant differences between metabolites related to performance. This seems likely to have occurred given the vast majority of PCs was higher in faster finishers but did not reach statistical significance. Second, our performance endpoint of finish time was unlikely to have been perfectly matched with level of physical fitness as highly physically fit individuals may have suffered injuries or non-fitness related setbacks which slowed their finish time. Third, information on our control subjects was limited to sex, race, age range and that they were “non-athletic”. The control subjects’ plasma samples were also obtained at a different time and location. It is worth mentioning that, even though our two study phases were an entire year apart from each other, we found consistent results with regards to PC levels both years, which was reassuring.

## 5. Conclusions

Our study found PC levels to be higher in faster 161-km ultramarathon finishers and higher in runners vs. non-athletic controls. Higher levels of a number of PCs are associated with faster 161-km ultramarathon finish times. Additionally, nearly all PCs were found to be significantly higher in ultramarathon runners, both before and after running an ultramarathon, than in non-athletic controls Future studies should attempt to elucidate the role of individual PCs in endurance sports performance, physical fitness and overall health.

## Figures and Tables

**Figure 1 sports-08-00044-f001:**
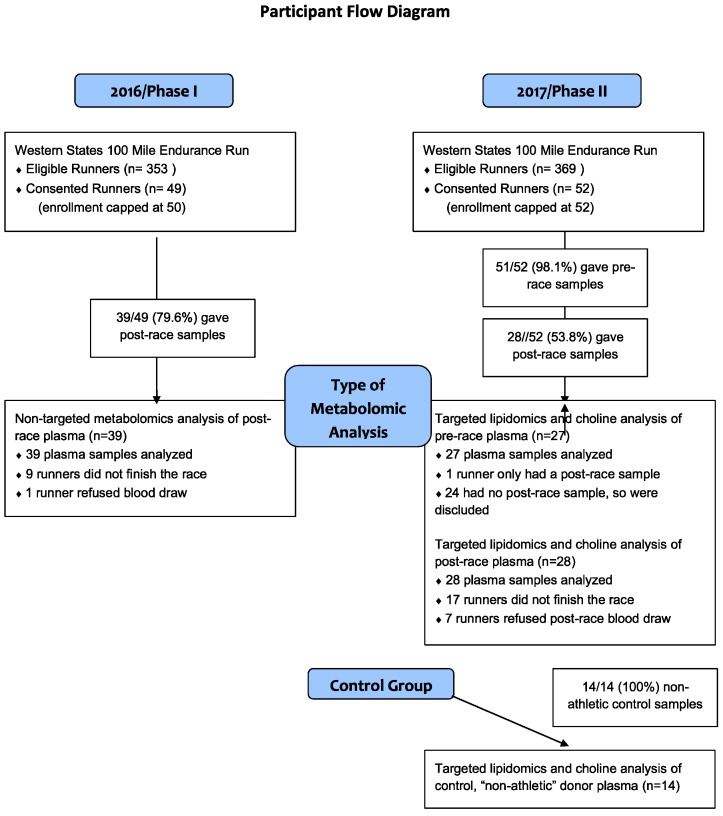
Participation and analyses performed in Phases I and II of the study.

**Figure 2 sports-08-00044-f002:**
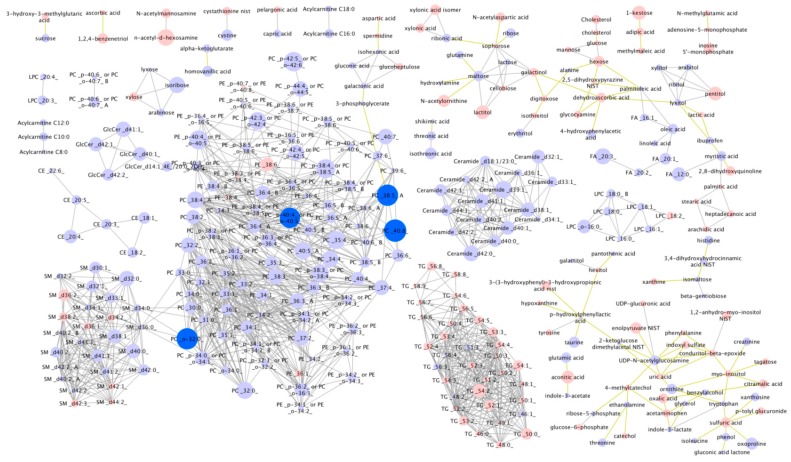
Individual post-race metabolite levels according to 161-km race finish time. Dark blue = higher in faster finishers (FDR < 0.05). Light blue = non-significantly higher in faster finishers. Pink = non-significantly lower in faster runners. Node size reflects the magnitude of the correlation using the correlation coefficient. Linear distances between objects are calculated using the Tanimoto or KEGG coefficients (tests of structural similarity).

**Figure 3 sports-08-00044-f003:**
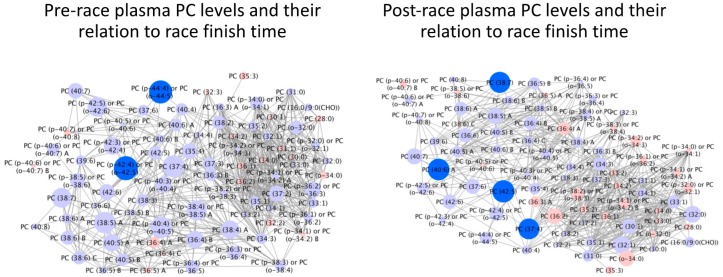
Individual pre- and post-race PC levels according to 161-km race finish time. Bright blue = higher in faster finishers (FDR < 0.05). Light blue = non-significantly higher in faster finishers. Pink = non-significantly lower in faster runners. Node size reflects the magnitude of correlation using correlation coefficient. Linear distances between objects are calculated using the Tanimoto or KEGG coefficients (tests of structural similarity).

**Figure 4 sports-08-00044-f004:**
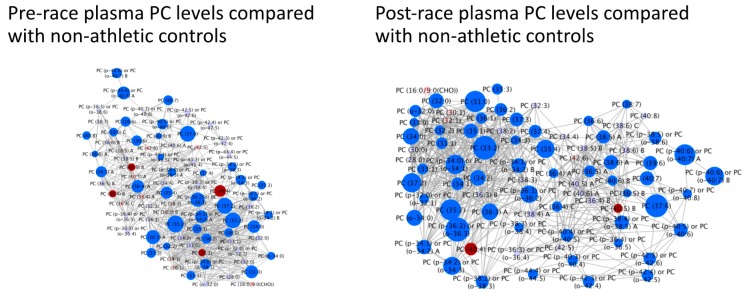
Individual pre- and post-race PC levels in runners compared with non-athletic controls. Bright blue = higher in runners than controls (FDR < 0.05). Red = lower in runners than controls (FDR < 0.05). Light blue = non-significantly higher in runners. Pink = non-significantly lower in runners. Node size reflects the magnitude of the correlation using the correlation coefficient. Linear distances between objects are calculated using the Tanimoto or KEGG coefficients (tests of structural similarity).

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
