# Peer review of "Ultramarathon Plasma Metabolomics: Phosphatidylcholine Levels Associated with Running Performance"

_sports, 2020, doi:10.3390/sports8040044_

Round 1

Reviewer 1 Report

An interesting study investigating plasma metabolomics in fast and slow ultramarathon finishers and in active and inactive populations. I have a few suggestions for the authors to improve the paper. The most important thing, I feel, is to add more detail to the introduction.

General comments:

The introduction seems incomplete. There doesn't seem to be any rationale for the study. You go straight from saying what the metabolome is, to saying your hypothesis. I appreciate that the first part of the study was non-targeted, but some description of some potentially important metabolites and the role they play would be useful.

Line 81. Why not have a one-to-one ratio for participants and controls? And why not try to age match, as well as sex match?

Figures 3 and 4 could each be two separate figures. That would make them easier to see.

Line 215. Could this limitation have been addressed in the design of the study by simply removing the cap of 50 participants?

Line 218. Could this limitation also have been addressed by obtaining blood samples from known subjects, rather than buying blood samples from unknown subjects?

Specific comments:

Line 89. Should this paragraph state that this is the 2017 data?

Lines 99-101. Unnecessary. 

Some reference to Figure 1 in the Methods section would be useful.

Line 125. Should this not be in the stats part of the Method section?

Lines 128 and 129. You've already stated this in the Method, so it isn't necessary here.

Line 170. Would it not be more appropriate to say "higher levels of certain phosphatidylcholines".

Line 174. I would give the author credit for the study, rather than their nationality or the nationality of their instituition. Same on line 213.

Line 204. Missing the name of the author.

Reviewer 2 Report

This paper aimed to identify plasma metabolites associated with superior endurance running performance. I think the paper needs a better structure to be published in Sports

The introduction is too brief. Authors should deepen in the scientific literature. In addition, which is the gap in literature? This needs a better justification. This section should be improved substantially.

Methods section needs to be rewritten as a better structure. Moreover, authors should explain the running performance variables.

In discussion section, it would be interesting to include a new paragraph addressing the practical implications.

Definitively, this paper needs to reformulate some sections to be published as a article in this journal.

Round 2

Reviewer 2 Report

I congratulate authors for the effort in making changes.

My unique observation is about the performance endpoint explaining the characteristics of the 161-km race.
